# Automated, high-dimensional evaluation of physiological aging and resilience in outbred mice

**Zhenghao Chen, Anil Raj, GV Prateek, Andrea Di Francesco, Justin Liu, Brice E Keyes, Ganesh Kolumam, Vladimir Jojic, Adam Freund\***

Calico Life Sciences LLC, South San Francisco, South San Francisco, United States

**Abstract** Behavior and physiology are essential readouts in many studies but have not benefited from the high-dimensional data revolution that has transformed molecular and cellular phenotyping. To address this, we developed an approach that combines commercially available automated phenotyping hardware with a systems biology analysis pipeline to generate a high-dimensional readout of mouse behavior/physiology, as well as intuitive and health-relevant summary statistics (resilience and biological age). We used this platform to longitudinally evaluate aging in hundreds of outbred mice across an age range from 3 months to 3.4 years. In contrast to the assumption that aging can only be measured at the limits of animal ability via challenge-based tasks, we observed widespread physiological and behavioral aging starting in early life. Using network connectivity analysis, we found that organism-level resilience exhibited an accelerating decline with age that was distinct from the trajectory of individual phenotypes. We developed a method, Combined Aging and Survival Prediction of Aging Rate (CASPAR), for jointly predicting chronological age and survival time and showed that the resulting model is able to predict both variables simultaneously, a behavior that is not captured by separate age and mortality prediction models. This study provides a uniquely high-resolution view of physiological aging in mice and demonstrates that systems-level analysis of physiology provides insights not captured by individual phenotypes. The approach described here allows aging, and other processes that affect behavior and physiology, to be studied with improved throughput, resolution, and phenotypic scope.

**\*For correspondence:**
adam@afbio.co

## Editor's evaluation

Chen et al., develop a comprehensive platform to score aging-dependent changes in mouse physiology and behavior using a multi-dimensional longitudinal phenotyping approach. Their thorough data collection and analysis reveals a diversity of trajectories in aging-related physiological and behavioral changes and helps disentangle biological aging from chronological aging, providing a pioneering reference for future studies aimed at large-scale aging multi-dimensional phenotyping.

## Introduction

The laboratory mouse is commonly used to study aging and test putative aging interventions (***Ackert-Bicknell et al., 2015***; ***Yuan et al., 2011***). While many studies have used lifespan extension as an endpoint, lifespan in mice is not necessarily the best criterion for assessing the efficacy of aging interventions or for evaluating their potential translation to humans; measuring health is an important additional source of information (***Hansen and Kennedy, 2016***; ***Fischer et al., 2016***). Health is a multi-parameter, organism-level state, and as multiple aspects of health decline with age, a sophisticated

assessment of aging and putative aging interventions should evaluate as many organism-level aspects of health as possible (*Freund, 2019*).

Recognizing this, the field has developed assays to evaluate different aspects of animal behavior, physiology, and function. Particular emphasis has been given to assays designed to measure cognitive, metabolic, and neuromuscular metrics, as well as body size and composition (*Huffman et al., 2016*; *Bellantuono et al., 2020*; *Sukoff Rizzo et al., 2018*; *Richardson et al., 2016*). However, many of these assays are labor- and time-intensive, thus the number that any one study can include is limited. Further, these assays often utilize a challenge, for example, performance on a difficult task, and due to the stress this places on the animals as well as changes in performance due to training, challenge-based assays have limited repeatability, reducing their utility for longitudinal analysis (*Huffman et al., 2016*). Additionally, investigators must pay attention to several confounding variables and potential artifacts such as motivation to execute the task, order of assays, and time allowed for recovery between assays (*Crabbe et al., 1999*; *Chesler et al., 2002b*; *McIlwain et al., 2001*). Lastly, many assays are highly operator-dependent, adversely impacting reproducibility (*Brown et al., 2018*; *Voelkl et al., 2020*). Thus, although the use of multiple challenge-based assays provides insight into fitness and health, this phenotyping strategy has serious limitations.

To develop an alternative approach, we utilized commercially available, automated monitoring cages that simultaneously and continuously record physiological and behavioral parameters (voluntary animal position, respiration, weight, food and water intake, and wheel activity) (*Brown et al., 2018*). This type of hardware is not new, but the data from these systems has been underutilized. Each of the sensors in the cage provides dense time series data, with time windows ranging from <1 s (photo-beam breaks) to 3 min (gas measurements). The most common analytical approach is to average this data into 12 hr time bins, that is, dark and light phase. Although useful for detecting large effects, more granular analyses provide more information. We hypothesized that, by generating hundreds of features from these raw data streams, insights into behavior and physiology could be uncovered via tools such as network modeling, which have previously been restricted to lower levels of biological organization. Additionally, because these measurements are acquired non-invasively during the course of daily living, observation can be continued for multiple days and repeated multiple times throughout an animal's life. Observation across multiple days provides important information about circadian stability and captures animal behavior during both the light and dark phases; this is difficult to accomplish using any other method (*Brown et al., 2018*). Repeated, longitudinal phenotyping provides major advantages for aging studies by allowing for baseline normalization (which increases statistical power), direct measurement of rates of change, predictive modeling of future outcomes, and parsing of survivorship bias (*Bellantuono et al., 2020*; *Zhang and Pincus, 2016*). Further, automated phenotyping removes confounders such as operator-induced variability, increasing data robustness (*Chesler et al., 2002a*). Lastly, the system can be readily scaled without compromising data continuity and without the need for training highly specialized staff.

We used this platform to assess aging in the diversity outbred (DO) population, a heterogeneous stock derived from the intercrossing of eight inbred founder strains, in which each mouse is genetically unique (*Churchill et al., 2012*). Widespread reliance on a small number of inbred mouse strains for preclinical studies raises questions about generalizability. Phenotype and intervention response are highly dependent on genetic background, meaning that the findings in one mouse strain may not generalize to other mouse strains, let alone to humans (*Voelkl et al., 2020*; *Sittig et al., 2016*; *Mandillo et al., 2008*; *Kafkafi et al., 2018*; *Tuttle et al., 2018*). This is a concern across many disease areas, including aging (*Liao et al., 2010*). In recent years, mouse resources with high genetic diversity and phenotypic variation have become available, and this diversity provides important advantages. First, it allows for genetic analysis of phenotypes, which can be used to validate phenotypic data based on measurable heritability, to better understand the correlation between traits, and, with sufficient sample size, to genetically map traits (*Svenson et al., 2012*; *Churchill et al., 2012*). Second, genetic diversity introduces inter-individual variation, which is useful when building networks based on phenotypic correlations. Third, and perhaps most importantly, genetic diversity introduces deliberate biological variation in order to increase the external validity of study findings (*Voelkl et al., 2020*).

We generated ~60 years of data across 415 female DO mice, from an age range of 6 weeks to 40.5 months (3.4 years). We utilized a staggered enrollment design in which mice were enrolled at different ages, between 3 and 24 months, to ensure that the dataset contains hundreds of runs for all

age groups up to 30+ months of age, providing consistent statistical power across an average mouse lifespan. To our knowledge, this represents the highest-resolution assessment of murine physiological aging to date: we examined animals monthly, whereas previous assessments of physiological aging sampled animals in ≥6-month intervals (*Ladiges et al., 2017*; *Fischer et al., 2016*; *Petr et al., 2021*). High-resolution phenotyping was manageable because our passive monitoring platform required only 20 hr of hands-on time per week from a single operator; as we and others have experienced, challenge-based phenotyping pipelines require so much hands-on time and often place such stress on the animals that they cannot be repeated frequently.

The platform detected hundreds of aging-related changes starting early in life, demonstrating that this passive monitoring approach has the sensitivity to quantify aging even in a highly variable strain background. Network analysis demonstrated that different behavioral and physiological clusters exhibited distinct aging dynamics. By employing analytical methods to measure overall network connectivity, we quantified resilience, an emergent property, and found that it exhibited an accelerating decline that was distinct from any individual phenotype. Finally, we develop a new method for determining biological age: a Combined Age and Survival Prediction of Aging Rate (CASPAR) model that is trained to simultaneously predict both chronological age and survival time. In summary, we demonstrate that automated phenotyping of murine behavior and physiology, when combined with high-dimensional analytics, identifies aspects of aging and resilience that have not been previously described and provides important advantages over existing approaches in terms of throughput, temporal resolution, and physiological scope. Our study builds upon pre-existing literature from other model organisms, particularly nematodes, demonstrating that passive, automated monitoring can be used to quantify multi-dimensional, organism-level aging (*Zhang et al., 2016*; *Le et al., 2020*; *Martineau et al., 2020*). This combined hardware/software platform can be replicated in any vivarium and used for in-depth characterization of behavior and physiology as well as intervention testing.

## Results

Many assessments of aging utilize a challenge, for example, a maze for cognitive function or a rotating rod for coordination, on the premise that measuring the limits of an animal's ability is the optimal way to detect aging-related changes (*Bellantuono et al., 2020*). We hypothesized that aging-related changes could also be detected without an overt challenge, by employing sufficiently sensitive and temporally dense measurements during the course of normal daily living, a concept analogous to measuring one's breathing rate after a brisk walk instead of after sprinting 100 m. To test this idea, we constructed a system of 64 semi-automated phenotyping cages that continuously measured multiple aspects of mouse physiology and behavior (*Figure 1A*). Cages record oxygen and carbon dioxide concentration, water vapor, weight of food and water hoppers, mouse body mass (when inside the sleeping chamber), running wheel rotations, and mouse position via a three-dimensional array of infrared beams. Gas measurements are acquired every 3 min because the time constant of the cage (rate at which air is replaced) only allows for meaningful changes in gas concentrations every ~3 min. Other measurements are acquired every second. Because food and water are readily available, mice can be monitored continuously, without disruption, for up to weeks at a time. We chose to monitor each mouse for 1 week per month (*Figure 1B*), as we reasoned that (1) a 7-day run would provide enough data for a robust and stable estimation of physiological/behavioral status and (2) monthly rotations would be sufficiently granular to capture the pace of aging in mice. This schedule allowed us to regularly monitor fourfold as many animals as we had cages, that is, 256. We placed the same animal in different cages for different runs, to mitigate any cage-specific measurement effects.

We initially enrolled 256 female DO mice (*Churchill et al., 2012*) split among four age groups (n = 64 at 7, 14, 21, and 25 months), and we enrolled younger (~3 months) animals in semi-regular waves as the older cohorts expired. Enrolling animals at different ages ensured that different ages were similarly represented and mitigated the loss of power that occurs with time due to mortality. Enrolling animals at different calendar dates ensured that age and calendar date were not perfectly correlated, attenuating the effect of any unknown, but potentially confounding, environmental effects such as seasonal changes or temperature drift over time. We utilized DO mice in order to avoid strain-specific results that would be less likely to generalize; however, as a practical necessity, we only examined female mice in this study. In total, we enrolled 415 animals, with an age distribution that provided large numbers of animals at both young and old ages (*Figure 1—figure supplement 1A*). The study

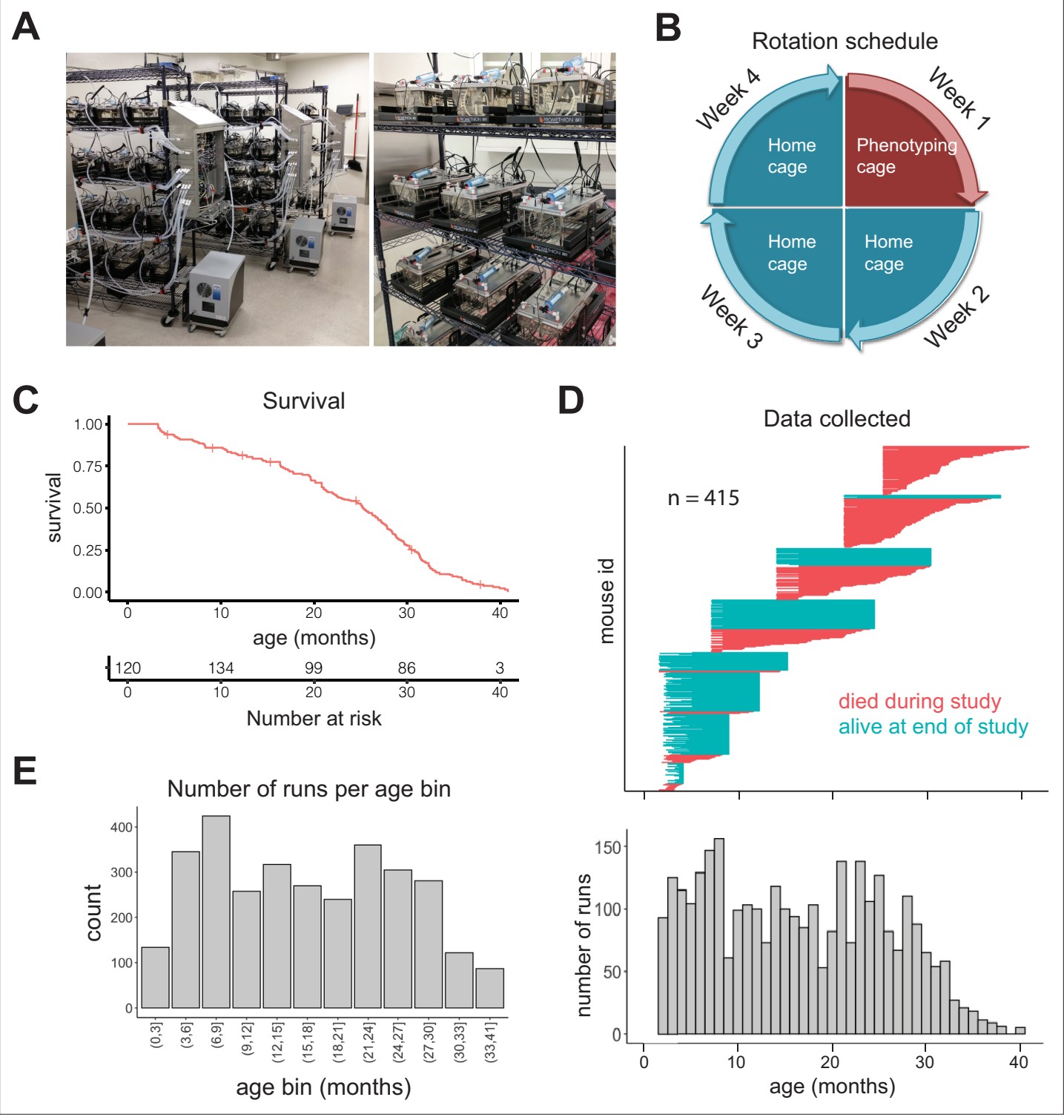

**Figure 1.** Study design. (**A**) Pictures of automated cage phenotyping system. (**B**) Rotation schedule for each animal; each animal was monitored 1 week per month. (**C**) Survival of cohort. Vertical lines indicate censorship (still alive) at end of study. (**D**) Top: age range over which data was collected for each animal; each row represents one animal. Bottom: number of runs collected at each month of age. (**E**) Number of 7-day runs collected for each 3-month age bin. Most downstream analysis was performed using these bins.

The online version of this article includes the following figure supplement(s) for figure 1:

**Figure supplement 1.** Study design.

was run for approximately 1.5 years, during which 212 mice died (51%) and 203 mice were censored at the end of the study (49%). This design is similar to human longitudinal studies in which subjects are enrolled at different ages and at different dates, but followed for similar lengths of time, leading to left-truncated and right-censored data. Accounting for truncation and censoring, DO mice had a median lifespan of 25 months (~2 years) and a maximal lifespan of 41 months (~3.4 years) (*Figure 1C*, *Figure 1—figure supplement 1B*). In total, after quality control filtering, we obtained data for 3143 7-day runs, that is, ~60 years of dense time series data, from an age range of 6 weeks to 40.5 months (3.4 years), with virtually all ages containing data from multiple cohorts (*Figure 1D*). For analysis of aging in the following sections, we grouped runs into 3-month age bins, providing a large number of runs (>200) in most bins (*Figure 1E*). Even with this staggered enrollment design, animals older than 33 months were rare; so for the oldest animals, we combined all runs for animals 33–41 months, for a total of 87 runs in the oldest age bin.

Because of their genetic diversity, DO mice are generally more variable than inbred strains. Indeed, we found a broad body mass distribution in this cohort, ranging from 15 to 60 g (*Figure 1—figure supplement 1C*). Given this heterogeneity, we sought evidence that the phenotyping cages, which were optimized for less diverse strains, were functioning properly and capturing physiologically meaningful information. In a large, multi-location analysis of indirect calorimetry data, the most consistent relationship was a positive correlation between body mass and energy expenditure (*Corrigan et al., 2020*). This positive relationship has been proposed as one indication of the technical success of an experiment (*Tschöp et al., 2011*). We compared body mass to energy expenditure (kcal/hr) in this dataset and saw a highly significant correlation (*Figure 1—figure supplement 1D*). As we already had body mass information, we were primarily interested in changes to energy expenditure and related parameters that were body mass independent, therefore for all gas-derived measurements except RQ ($VO_2$, $VCO_2$, $VH_2O$, and energy expenditure), because it is a ratio, we normalized for body mass via linear regression. This removed the positive correlation with body mass (*Figure 1—figure supplement 1E*). This normalization had the potential to induce spurious correlations between gas-derived measurements, so we compared the correlation coefficients between these measurements before and after body weight correction (*Figure 1—figure supplement 1F*). As expected, there was some correlation between most gas measurements prior to correction. This correlation increased slightly, but not appreciably, upon body mass correction (up to a 0.04 increase in R). We believe that the benefit of focusing on body mass-independent information outweighs the slight increase in correlation between gas measurements that we observed, so we chose to use body mass-corrected gas measurements for all subsequent analysis steps.

## Automated phenotyping identifies robust physiological changes at each stage of life

We developed a data processing pipeline that provides aggregated, quality-controlled data at various levels of resolution (*Figure 2A*). Initially, data are processed into 14 'base features' in 3 min bins: BodyMass (weight), Food (food intake), Water (water intake), WheelMeters (distance traveled on wheel), PedMeters (distance traveled in cage), AllMeters (PedMeters + fine motor movements), XBreak (number of photobeam breaks in the X axis), YBreak (number of photobeam breaks in the Y axis), ZBreak (number of photobeam breaks in the vertical axis), $VO_2$ (oxygen consumption), $VCO_2$ (carbon dioxide production), energy expenditure (kcal/hr), RQ (respiratory quotient – $VCO_2/VO_2$), $VH_2O$ (water vapor). After quality control, each run was split into 24 hr windows and those windows were averaged to generate the average 24 hr trace for each feature for each run. We then grouped runs into the 3-month age bins described above and examined the 14 base features for signs of aging-related change (*Figure 2—figure supplement 1A*). As expected, animals exhibited circadian patterns of behavior and physiology, with increased activity during the dark phase. Clear aging-related change were observed for most of the 14 base features, starting from the youngest ages we examined. One of the most striking changes we observed was a concave body mass trajectory: animals gained weight until 21–24 months of age, then progressively lost weight (*Figure 2B*). Other striking changes included a significant decline in wheel running, food intake, water intake, and a decline in mass-adjusted energy expenditure and rate of whole-body water loss ($VH_2O$) (*Figure 2B*). Consistent with these observations, it has been previously reported that body weight declines in older mice despite a decrease in energy expenditure (*Petr et al., 2021*). Not all base features declined with age – ambulatory activity

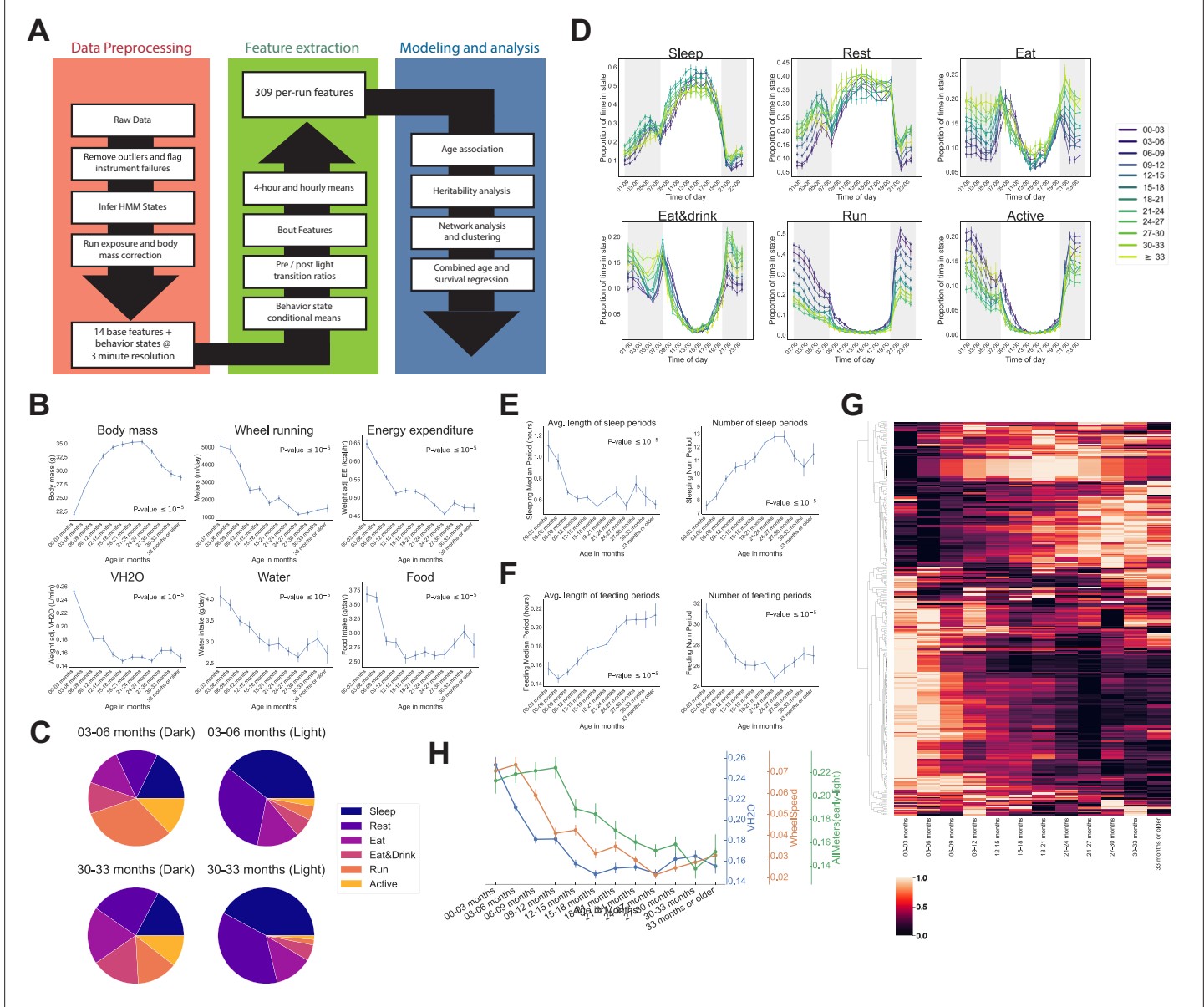

**Figure 2.** Automated phenotyping identifies physiological changes at each stage of life. (**A**) Diagram of data processing pipeline. Three-min raw data from each run are fed through a quality control pipeline to remove data from broken and/or miscalibrated sensors. Data is then analyzed by a robust hidden Markov model (HMM) to assign each 3 min window to one of six states. Gas measurements are then adjusted for body mass. Data are then aggregated in a variety of ways for feature extraction: by time window (1 or 4 hr windows), by periods of behavior (e.g., number of sleeping periods), as a ratio of values from before and after a light transition (e.g., RQ pre-lights on/RQ post-lights on), and by state (e.g., VCO₂ while running). Finally, data are used for modeling and analysis. (**B**) Effect of age on six of the base features of the phenotyping cages. Data were averaged for each run and then averaged across runs within each age bin. (**C**) Average percent time spent in each state for young (3–6 months) and old (30–33 months) animals, split by dark/light phase. (**D**) Average occupancy of each HMM state at each hour of the day. Timepoints represent the preceding hour, for example, the 7 pm timepoint includes data from 6 to 7 pm. (**E**) Effect of age on average duration and daily number of sleeping periods. (**F**) Effect of age on average duration and daily number of feeding periods. (**G**) Heatmap of aging trajectories. Rows are features (n = 309), columns are monthly age bins. Each feature was normalized and scaled across age bins from lowest value (0) to highest value (1). (**H**) Examples of four features that decrease with age, with different ages of onset. All error bars are SEM.

The online version of this article includes the following figure supplement(s) for figure 2:

**Figure supplement 1.** Automated phenotyping identifies robust physiological changes at each stage of life.

**Figure supplement 2.** Effect of age on state-conditioned features.

**Figure supplement 3.** Genetic analysis of features.

(PedMeters, AllMeters, and YBreaks) tended to increase with age (*Figure 2—figure supplement 1A*). None of these changes mirrored the concave trajectory of body mass, demonstrating that multiple aging-related changes to physiology and behavior are not driven solely by weight changes.

Aging-related changes could arise from altered behavior and/or physiology. For example, reduced distance run on the wheel could arise from reduced running speed or reduced time spent running, which might arise from different underlying mechanisms. To quantify behavior, we fit a robust hidden Markov model (HMM) that uses base features of the phenotyping cages, excluding body mass, to classify each 3 min time window into one of six states. A mouse can perform many different behaviors in 3 min, meaning that each 'state' is a combination of individual behaviors. Each state was characterized by a distinct combination of base features (*Figure 2—figure supplement 1B*), and we named the states Sleep, Rest, Eat, Eat&Drink, Run, and Active based on our qualitative interpretation of those patterns. Active, Eat, and Eat&Drink in particular are mixed states, characterized by different combinations of eating, drinking, movement around the cage, and running on the wheel. As measured by energy expenditure, Sleep, Rest, and Eat are lower activity states, whereas Eat&Drink, Run, and Active are higher activity states. As expected, mice organized their behavior around the light cycle, with sharp transitions at lights on/off. They spent the vast majority of the light phase in lower activity states, whereas high activity state occupancy increased in the dark phase (*Figure 2—figure supplement 1C*). There were differences in overall state occupancy between young (3–6 months) and old (30–33 months) mice; compared to young mice, old mice spent more time in low activity states (*Figure 2C*). This was true during both light and dark phases, though it was particularly obvious during the dark phase, where old mice spent far less time Running or Active than young mice and more time in the other four, lower activity, states.

Circadian averages are a coarse description of animal behavior, and the temporal resolution of the data allows for more thorough examination. At the hourly (*Figure 2D*) and 4-hr level (*Figure 2—figure supplement 1D*), we observed a number of interesting patterns of mouse behavior. Animals exhibited a sharp increase in activity when the lights turned off, reaching their highest activity in the early part of the dark phase, about 2 hr after lights off. During this time, young animals spent nearly 70% of their time Running or Active. Older animals also exhibited their highest activity during this time, but their peak activity was much lower than young mice, and their transition to high activity when the lights initially turned off was slower. Running and Active then generally declined in frequency throughout the rest of the dark phase. Mid-level activity states, that is, Eat and Eat&Drink, exhibited a different pattern: animals spent most of their time in these states near the light/dark transitions. At the end of the dark phase, in particular, this corresponded to a large increase in food and water intake (see *Figure 2—figure supplement 1A*), likely indicating that the animals were 'stocking up' before the low-activity light phase. Prior to this large increase in food and water intake, animals reached a dark phase minimum for food/water intake and a maximum for Sleep and Rest, that is, they took a nap. Napping behavior exhibited different aging dynamics than the decline in high activity noted above; whereas Run and Active declined until ~24 months of age, napping increased in frequency until ~18 months and then stabilized (see Rest during dark phase time windows, *Figure 2—figure supplement 1D*).

We also saw evidence of circadian phase shifts with age, though not in the manner we expected. We assumed older animals would show less entrainment to the light cycle as they aged, indicative of circadian deregulation. However, we found that old animals were actually more entrained to the light cycle than young animals: young animals reached their peak Eat and Eat&Drink occupancy 1 hr after the lights turned on, during what is generally considered the low activity, 'sleep' phase of the circadian cycle (*Figure 2D*). In contrast, older animals reached their peak Eat and Eat&Drink occupancy in the hour prior to lights on, then quickly transitioned to Sleep and Rest once the lights turned on. The reason for this is unclear – it may be that young mice organize their circadian behavior around wheel running instead of eating, that is, they spend so much of the dark phase running that eating necessarily occurs during the light phase (*Edgar and Dement, 1991*; *Yamanaka et al., 2013*; *Yasumoto et al., 2015*). As wheel running declines with age, there is more time during the dark phase for eating, so peak Eat&Drink occupancy shifts (*Valentinuzzi et al., 1997*). Whatever the causal chain, this increased entrainment of feeding behavior to the light cycle with age represents an aging-related circadian shift that, to our knowledge, has not been previously described.

Irrespective of age, all mice reached a peak Sleep occupancy of ~50% in the middle of the light phase, and overall sleep duration did not change with age (*Supplementary file 1*). However, the

temporal granularity of the data allowed for an assessment of sleep fragmentation, that is, the duration of continuous periods of sleep, for which we did observed an aging effect. In young mice, periods of continuous sleep averaged 1.1 hr in duration; this declined quickly with age, plateauing at ~0.6 hr by 9–12 months of age, representing a 30 min reduction in the duration of an average sleep period (*Figure 2E*). At the same time, number of sleeping periods increased with age, representing increased sleep fragmentation. Increased behavioral fragmentation was not observed for all states: in contrast, the average duration of feeding periods increased with age while number of feeding periods decreased (*Figure 2F*), representing a consolidation of feeding behavior. Increased food spilling has been reported in aging mice (*Starr and Saito, 2012*); this may result in old mice requiring more time than young mice to ingest an equal quantity of food and may explain the extended duration of feeding periods with age.

The above analyses showed that murine aging is characterized by marked changes in behavioral patterns. The aging-related changes in base features noted above (*Figure 2—figure supplement 1A*) are thus affected by changes in both behavior and underlying physiology. To decouple these contributions, we computed averages of each of the 14 base features conditioned on the HMM state and compared how these changed with age (*Figure 2—figure supplement 2A*). We included two additional features – WheelSpeed and PedSpeed – which are the speed of wheel running and pedestrian locomotion, respectively, for a total of 16 base features. Needless to say, these two rate metrics are the most physiologically informative when the animal is actively running or walking, and the plots show that calculated running and walking speed during other states (e.g., Sleep) is under-estimated because the animal spends considerable time not running or walking during these states.

Examining energy expenditure while mice were resting demonstrated that the decline in overall mass-adjusted energy expenditure was, at least partly, a decline in resting metabolic rate – mice exhibited a decline in low-activity-state (Rest and Sleep) energy expenditure with age (*Figure 2— figure supplement 2B*). Energy expenditure during high activity states (Run and Active) was also decreased, indicating reduced activity intensity. Consistent with this, animals ran more slowly as they aged throughout most of life (*Figure 2—figure supplement 2C*). Animals also exhibited a aging-related decline in mass-adjusted $VH_2O$, that is, whole-body water loss rate, during all states, including Rest and Sleep (*Figure 2—figure supplement 2D*), suggesting that this is a consequence of altered internal physiology rather than altered behavior. Interestingly, the decline in water loss rate precedes the decline in overall water intake (see *Figure 2A*), so reduced water intake does not explain this effect; it may instead represent aging-related changes in body composition, as lean and fat mass retain different amounts of water (*Vu et al., 2017*).

Many other features changed with age as well, so in order to obtain a holistic view of aging patterns, we generated 309 features from the raw data using a number of different approaches, including those mentioned above: (1) the per-run average of each of the 16 base features noted above + average occupancy of the 6 HMM states ($16 + 6 = 22$ features), (2) the 16 base features conditioned on each state ($16 \times 6 = 96$ features), (3) average values of the 16 base features and 6 HMM states during each of six 4-hr time bins ($[(16 + 6)] \times 6 = 132$ features), (4) the ratio of values from before and after a light transition for each of the 16 base features and 6 HMM states ($(16 + 6) \times 2 = 44$ features), and (5) period number, average and maximum period length, average and maximum time interval between periods for eating, sleeping, and high-activity ($5 \times 3 = 15$ features) (see *Figure 2A*, green box). We chose these different aggregation approaches because we reasoned they had the highest chance of capturing unique aspects of physiology, though additional features may be informative. For example, the length of sleeping periods might show a different aging pattern than overall sleep amount, and the ratio of pedestrian locomotion before and after a light transition might provide different physiological information than pedestrian locomotion at a particular time of day.

After multiple hypothesis correction, 244 features (79%) significantly changed with age (*Supplementary file 1*). Features exhibited a variety of trajectories and ages of onset (*Figure 2G*), many of which were distinct from the base features, demonstrating that feature engineering uncovered additional physiological information. We chose a selection of three such features to highlight the different ages of onset we observe (*Figure 2H*). This diversity of aging patterns shows that (1) passive phenotyping has the sensitivity to detect aging across the entire lifespan and (2) physiological aging is more complex than simple linear trajectories and thus both dense time sampling and multi-dimensional phenotyping are needed to accurately capture the many different domains of physiological aging.

## Physiological features are influenced by genetics

Each animal in the DO population is genetically unique, allowing for estimation of the heritability and genetic architecture of traits. This analysis serves as a test of the biological validity of the different aggregation approaches noted above, because only approaches that capture real biological signal should exhibit significant heritability. The number of mice in this study is underpowered for estimation of low heritabilities (<10%) and for genetic mapping (*Gatti et al., 2014*), so we chose to focus on genome-wide measures of heritability and genetic correlations, rather than mapping individual loci for each trait. We genotyped all mice in the study, determined the kinship matrix, and calculated the heritability of each of of the ~300 features mentioned above, using all age groups (*Figure 2— figure supplement 3A*), excluding period features because these were not calculated across all base features. As expected due to low power, many traits exhibited heritabilities that were indistinguishable from zero. Also as expected, body mass exhibited high heritability (~30%); body mass in DO mice has previously been shown to be highly heritable (*Wright et al., 2020*) and is largely invariant to state or time of day, hence high heritability across aggregation methods. Importantly, at least some features derived from each of the aggregation approaches showed significant heritability, and in many cases these state- or time-conditioned base features showed higher heritability than the same base feature averaged across the entire run (the 'overall' feature). For example, heritability of X and Y breaks during the Eat&Drink state was higher than overall X and Y breaks. This suggests that the different aggregation approaches capture biological information that is distinct from overall averages.

After removing all body mass-derived features except for overall body mass in order to avoid redundancy, 33 features exhibited significant nonzero heritability (*Figure 2—figure supplement 3B*). The four most heritable features, with heritabilities >30%, involved pedestrian locomotion (AllMeters, PedMeters, PedSpeed, Xbreaks, or YBreaks). In total, 17/33 (52%) of significantly heritable features involved pedestrian locomotion, suggesting that animal movement is an aspect of behavior with a strong genetic component. In addition, we detected significant heritability for several HMM states during different times of day (e.g., Eat&Drink, Run, and Rest), demonstrating that this approach to cataloging mouse behavior captures genetically influenced biological processes. Interestingly, aside from several energy expenditure-based features, the body mass-conditioned gas measurements mostly lacked significant heritability (or at least, the heritability was below our ability to detect), suggesting that respiration, once accounting for body mass, is mostly influenced by environment rather than genetics.

In order to measure the degree to which genetic variants had similar effects on multiple features, we calculated the genetic and phenotypic correlations for the 33 features with significant heritability (*Figure 2—figure supplement 3C*). Genetic correlation measures the similarity of the genetic effects between two traits, for example, a genetic correlation equal to 1 means that every variant that affects the first trait has a proportional effect on the second trait. We observed a large number of positive and negative genetic correlations; in fact, the genetic correlations were, on average, stronger than phenotypic correlations (median genetic correlation = 0.22, median phenotypic correlation = 0.16), which is reasonable if the different environmental contributions to a pair of phenotypes are mostly uncorrelated noise. We applied unsupervised hierarchical clustering to identify clusters of features based on their genetic correlation. A few examples serve to illustrate the information in this plot, starting with the most trivial: variants that increased Rest frequency during a particular time window (late-dark) decreased the time spent in other states in that same window (not surprising, since state occupancy is a zero sum game). Slightly less obviously, Running and energy expenditure features form a cluster; variants that increase Running time increase energy expenditure. Variants that increased pedestrian locomotion in one state or time window tended to increase pedestrian locomotion in other states and time windows and formed several large clusters, suggesting the genetic contribution to pedestrian locomotion affects an overall propensity to move around, rather than influencing movement during any particular time of day. The features most genetically correlated to BodyMass were Eat&Drink occupancy during the late-light phase and X/Y Breaks during the Eat&Drink state, providing quantitative evidence that genetic variants which alter the timing and pattern of eating and drinking have a significant effect on body mass.

It was visually apparent that the genetic and phenotypic correlations for these 33 features exhibited a similar pattern, and to quantify this, we compared the genetic and phenotypic correlations for all pairs of these 33 features on a scatterplot and calculated the correlation between the two analyses

(*Figure 2—figure supplement 3D*). Genetic and phenotypic correlations were highly correlated across features ($R^2 = 0.9$, p $= 3.12 \times 10^{-196}$), suggesting that, for many trait pairs, the phenotypic correlation is due to shared genetics. Further, the high degree of similarity between genetic and phenotypic correlations suggest that, in future studies, additional genetic mapping power could be attained by analyzing phenotypes jointly.

## Quantifying relationships between physiological and behavioral features demonstrates a decline in resilience with age

Thus far we had examined how individual features change with age, but because the interplay between physiological domains is hypothesized to be a primary determinant of system resilience (*Cohen, 2016*), we were also interested in the connectivity between these features and how that connectivity changes with age. The strong genetic and phenotypic correlations noted above motivated us to explore this in more detail. A unique feature of this dataset is that all features were measured simultaneously in all runs, which allowed us to quantify the covariance between features, that is, how each feature is correlated with every other feature across runs. More precisely, we fit a graphical model (a semi-parametric Gaussian copula model) using sparse precision matrix estimation methods and visualized the model as a force-directed network (*Figure 3A*). We determined the number of clusters and clustering of features via consensus clustering, which resulted in 22 clusters. Some of these clusters were defined by the physical sensor used to generate the data, for example, VH$_2$O and CO$_2$ production, but most were comprised of features from multiple sensors. For example, beam break features did not form a single cluster, nor did all food, water, or gas features. Instead, clusters were defined by combinations of behaviors and aggregation methods: cluster 16 represents overall metabolic rate; cluster 12 represents food and water intake in a specific time window (the light phase); and cluster 13 represents the ratio of activity before/after a light transition, etc. (*Supplementary file 2*). Ratio-based features, state-conditioned features, and time-of-day conditioned features often clustered separately from one another; this demonstrates that different data aggregation approaches convey different information. Furthermore, this suggests that nodes within a cluster share similar physiological underpinnings rather than simply similar sensor hardware.

Some clusters were tightly integrated with other clusters, whereas others exhibited sparse connectivity, demonstrating their independence from other physiological domains. To better visualize this, we generated a chord diagram in which each cluster is placed around a circle, with chords demonstrating connectivity between nodes from different clusters (*Figure 3B*). The radius of each cluster arc is proportional to its overall connectivity to other clusters, thus larger clusters indicate more connections (not the number of nodes in the cluster). One of the most well-defined and least-connected clusters was body mass, as demonstrated by its small arc radius: all body mass metrics formed a tight cluster that was largely independent of other clusters. The tight clustering between mass metrics arises because body mass is mostly invariant with respect to behavior and time of day, so different body mass features from the same run are highly correlated to one another. The independence from other clusters arises partly because, as noted above, gas-derived measurements other than RQ (VO$_2$, VCO$_2$, kcal/hr, VH$_2$O) were adjusted for body mass. However, body mass was also largely independent of features that were not mass-adjusted, such as wheel, photobeam, food, and water features. The low connectivity of body mass to other phenotype clusters demonstrates that variance in body mass does not drive all, or even most, of the variance in other features in this dataset. This conclusion is further supported by the observation that body mass exhibits a clear concave trajectory with age, whereas most other features do not. In contrast, metabolic rate and high activity pattern were the most densely connected clusters, indicating that these physiological characteristics impact (or are impacted by) many other physiological domains.

We next sought to understand whether network connectivity changed with age, as a readout of system resilience. Resilience refers to the ability of a system to maintain function in the face of change. This is a broad concept that is unlikely to be fully captured by a single number, and multiple approaches to measuring organism-level resilience have been proposed (*Huffman et al., 2016*). Here, we propose a metric that is based on the relationship between physiological features. In a resilient system, a small disruption in one physiological domain will have limited system-wide effect, whereas in a less resilient system, the same perturbation will affect more of the network. Thus, in less resilient systems, individual subsystems (nodes) are sensitive to the fluctuations of other subsystems, leading to a rising

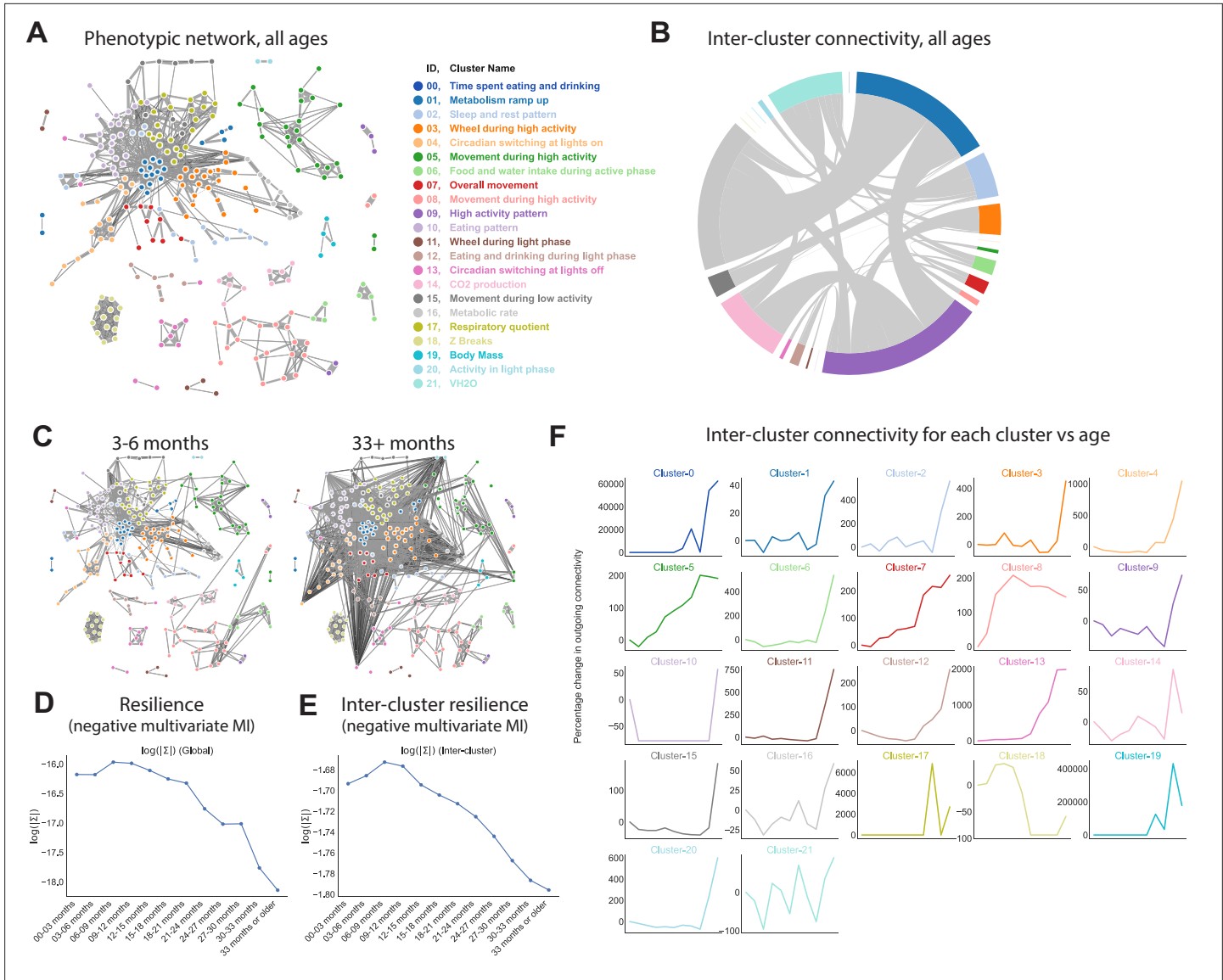

**Figure 3.** Quantifying relationships between physiological and behavioral features demonstrates a decline in resilience with age. (**A**) Force-directed network diagram of all features across all runs. Nodes represent individual per-run features, edges reflect the regularized covariance between two features (effectively, the correlation between two features after accounting for all other features). Increased edge thickness indicates increased covariance. Colors represent predicted clusters. (**B**) Chord diagram of connectivity between clusters across all runs. Colors are as in (**A**). (**C**) Network connectivity at young and old age. Edges calculated for each age bin independently, with position of nodes held constant. (**D**) Effect of age on resilience, that is, the negative multivariate mutual information (MI) of the age-specific graphical model. A lower value indicates stronger connection between features and a lower diversity of states occupied by animals within that age bin. (**E**) Effect of age on inter-cluster resilience. Resilience calculated from age-specific sub-networks built from exemplar features, one from each cluster. (**F**) Effect of age on inter-cluster connectivity. Connectivity value represents the sum of all edges connecting cluster nodes to nodes in other clusters, that is, the same metric as in (**B**), plotted across age groups.

The online version of this article includes the following figure supplement(s) for figure 3:

**Figure supplement 1.** Physiological and behavioral network models.

correlation between subsystems (*Scheffer et al., 2018*). For a graphical network in which correlation between nodes is visualized by edge weight, this manifests as increased network connectivity. Conversely, in a resilient system, because a small disruption in one node only affects a small number of other nodes, the average correlation between nodes is low, leading to lower network connectivity. Keeping node positions constant, we calculated network edges for each age bin (*Figure 3C*, *Figure 3—figure supplement 1A*). Visual inspection of these per-age networks from 3 to 6 months to 33+ months suggested increasing connectivity with time, suggesting decreasing system resilience.

To quantify network connectivity directly, we calculated the copula entropy/negative multivariate mutual information (MI) (*Singh and Póczos, 2017*) of the graphical model for each age bin. Multivariate MI is a measurement of how much information can be determined about the network as a whole from a subset of nodes. In one extreme case where all nodes are perfectly correlated (low resilience), knowing the value of a single node allows one to perfectly predict the value of all other nodes. Such a system has zero degrees of freedom and thus high MI. In the other extreme, nodes are completely independent (high resilience), provide no information about one another, and all must be measured to achieve a full description of the system – low MI. Thus, the negative of MI directly corresponds to a more rigid network and is a quantitative measurement of system resilience. Resilience (negative MI) exhibited a striking decrease with age, with an accelerating rate of decline; this pattern qualitatively reflects the accelerating failure rate associated with aging better than the semi-linear or early-life dropoff we observed with many individual parameters in this study (*Figure 3D*). Thus, resilience may be more accurate representations of physiological aging than are individual features.

To better understand what network-level changes were driving this decline in resilience with age, we examined both inter-cluster connectivity as well as intra-cluster connectivity, that is, we examined the relationship between physiological domains as well as the relationship between related features within physiological domains. To quantify inter-cluster connectivity, we represented each cluster by the exemplar feature that most reflected the behavior of that cluster (as measured by median consensus with other features in the same cluster). We then re-built networks using only those exemplars and calculated resilience. Like overall resilience, inter-cluster resilience exhibited a striking decrease with age (*Figure 3E*), suggesting the connectivity between subsystems increases with age. This effect was not merely driven by a few outlying clusters; increased connectivity with age was observed for the majority of clusters (*Figure 3F*). We also examined intra-cluster resilience, by measuring the negative MI between nodes within each cluster (*Figure 3—figure supplement 1B*). Most clusters showed a decline in resilience with age (increasing connectivity), but trajectories varied: some clusters exhibited a nearly linear decline in resilience, whereas others exhibited a late-life acceleration, perhaps indicating that certain domains of physiology are more prone to a loss of resilience with age than others.

## Age and time to death can be simultaneously predicted from physiological data

An increasingly common approach when dealing with multi-dimensional aging data is to train models that predict relevant endpoints (so-called aging 'clocks'). This approach was popularized using DNA methylation data (*Horvath, 2013*), and more recent reports use other data modalities including physiological measurements (*Schultz et al., 2020*). These studies have developed models that predict either chronological age or mortality risk separately, but these separate approaches are limited because neither chronological age nor mortality risk fully encapsulates an individual's multi-system, aging-related health status, that is, biological age.

To address this, we developed an aging rate regression model in which biological age is determined from a combination of chronological age and health status (in this case, time to death, though other health proxies such as a frailty score could be used). The model includes a hyperparameter that allows for tuning of the relative weighting of chronological age and time to death, allowing us to generate models with different behaviors. More specifically, this hyperparameter (denoted $\sigma_\beta$) quantifies our belief that different individuals age at different rates. If a ground truth measurement of individual aging rates existed, this hyperparameter could be measured empirically. Unfortunately, there remains no agreed-upon definition of biological age and no such ground truth is available. Therefore, here we explore model behavior under several different values of $\sigma_\beta$. A low value of $\sigma_\beta$ causes the model to assume that all individuals age at similar rates, meaning that the biological age of individuals of the same chronological age should be similar. In this case, model training heavily weighs chronological age, and the resulting model approximates a standard age clock model. Conversely, a high value of $\sigma_\beta$ causes the model to assume that individuals can age at different rates, and thus model training disregards chronological age, instead emphasizing health status (time to death), and the resultant model approximates a standard accelerated failure time model. Neither chronological age nor time to death are perfect representations of aging rate, and they are not particularly well correlated with one another (*Figure 4—figure supplement 1A*), thus optimizing the prediction of one necessarily reduces performance for the other, resulting in a tunable tension in model behavior and the ability to explore

intermediate states that may avoid overfitting to either of these imperfect biological age surrogates. Because this framework utilizes both chronological age and survival time as outcome variables, we name this approach the "Combined Age and Survival Prediction of Aging Rate", or CASPAR.

To understand how this trade-off behavior between chronological age and time to death manifests, and to determine if there is an acceptable middle ground, we trained models using different values of $\sigma_\beta$. For each model, we trained on the same 15 different random splits of the data and report the average performance of the model on the held-out mice. In each split, we allocated 90% of mice to training and 10% of mice to testing, with full animal holdouts (i.e., all runs of the same animal were either entirely in the training set or entirely in the test set). Note that because the model utilizes time to death, we only included the mice that died during the course of the study, approximately half the mice in the overall dataset. Leveraging the longitudinal nature of our dataset to improve the robustness of the model to run-specific effects, during model training we assume that the aging rate of an animal is constant over time, that is, that the observed aging rate of the same animal at different ages is modeled by a latent common aging rate plus a standard normal noise term.

At the lowest values of $\sigma_\beta$ we tested (e.g., $\sigma_\beta = 0.1$, heavy weighting of chronological age), when assessed against the test sets, the model's scores were highly correlated to chronological age, with a mean correlation across the 15 data splits of 0.77 (*Figure 4A*). However, as expected, this model performed poorly on time to death prediction, with a mean correlation between model scores and time to death of 0.33 (11% variance explained). Not surprisingly, in this case the model barely performs better than true chronological age, which explains about 8% of the variance in time to death (p < 1e − 4) (*Figure 4—figure supplement 1A*). This behavior represented one extreme of model performance, as the correlations for age and time to death were stable for all values of $\sigma_\beta \leq 1$ (*Figure 4B*).

In contrast, at the highest value of $\sigma_\beta$ (e.g., $\sigma_\beta = 100$, heavy weighting of time to death), the model's scores were better correlated to actual time to death with a mean correlation of 0.47 (*Figure 4A*). As expected, correlation with the chronological age was lower in this model, dropping to a correlation of 0.58. This behavior represented the other extreme of model performance, as the correlations for age and time to death were stable for all values of $\sigma_\beta \geq 30$ (*Figure 4B*). All 15 training/test splits of the data led to similar model performance, demonstrating that model performance is robust to the specific animals allocated to the training and test sets (*Figure 4B*).

Interestingly, the trade-off between age and time to death prediction was not linear – intermediate values of $\sigma_\beta$ improved prediction in one dimension without an equivalent loss of performance in the other dimension (*Figure 4A*). To better visualize this, we chose three values of $\sigma_\beta$ to represent three different regimes of the CASPAR model: the chronological age model ($\sigma_\beta = 0.1$), the time to death model ($\sigma_\beta = 100$), and the hybrid model, which represents a roughly equivalent trade-off between age and time to death ($\sigma_\beta = 3$). For each model class, we plotted actual age and actual time to death versus predicted age for each run in the test set, for each of the 15 models trained on different data splits (*Figure 4C*). As noted earlier, the chronological age model predicted age quite well ($R^2 = 0.60$, p < 1e − 4) but did not predict time to death much better than chronological age itself ($R^2 = 0.11$, $p = 0.001$), whereas the time to death model predicted time to death relatively well ($R^2 = 0.22$, $p < 1e − 4$) but chronological age relatively poorly ($R^2 = 0.34$, $p < 1e − 4$). Interestingly, the hybrid model performed relatively well on both tasks, predicting chronological age with an $R^2 = 0.50$ ($p < 1e − 4$) and predicting time to death with an $R^2 = 0.20$ ($p < 1e − 4$), which was substantially better than the true chronological age of the animal and nearly as good as the time to death model itself. This balanced prediction of both dimensions suggests that the hybrid model may represent a qualitatively different approach to quantifying health-relevant biological age than either age or survival prediction alone.

We examined the models to understand which features were most informative. For all three model classes (age, time to death, and hybrid), the top 20 most informative features came from multiple clusters and sensors (gas, photobeams, wheel, body mass) and aggregation methods (state-conditioned, time of day, lights on/off ratio) (*Figure 4—figure supplement 1B*, *Figure 4—figure supplement 1C*). No single cluster or sensor dominated, demonstrating the importance of a high-dimensional feature set. The features that informed each of the model classes were partially overlapping (*Figure 4D*). Of the top 20 features from each model, 7 were shared across all three (*Figure 4—figure supplement 1C*). Perhaps unsurprisingly, these include major aspects of physiology – overall size (BodyMass),

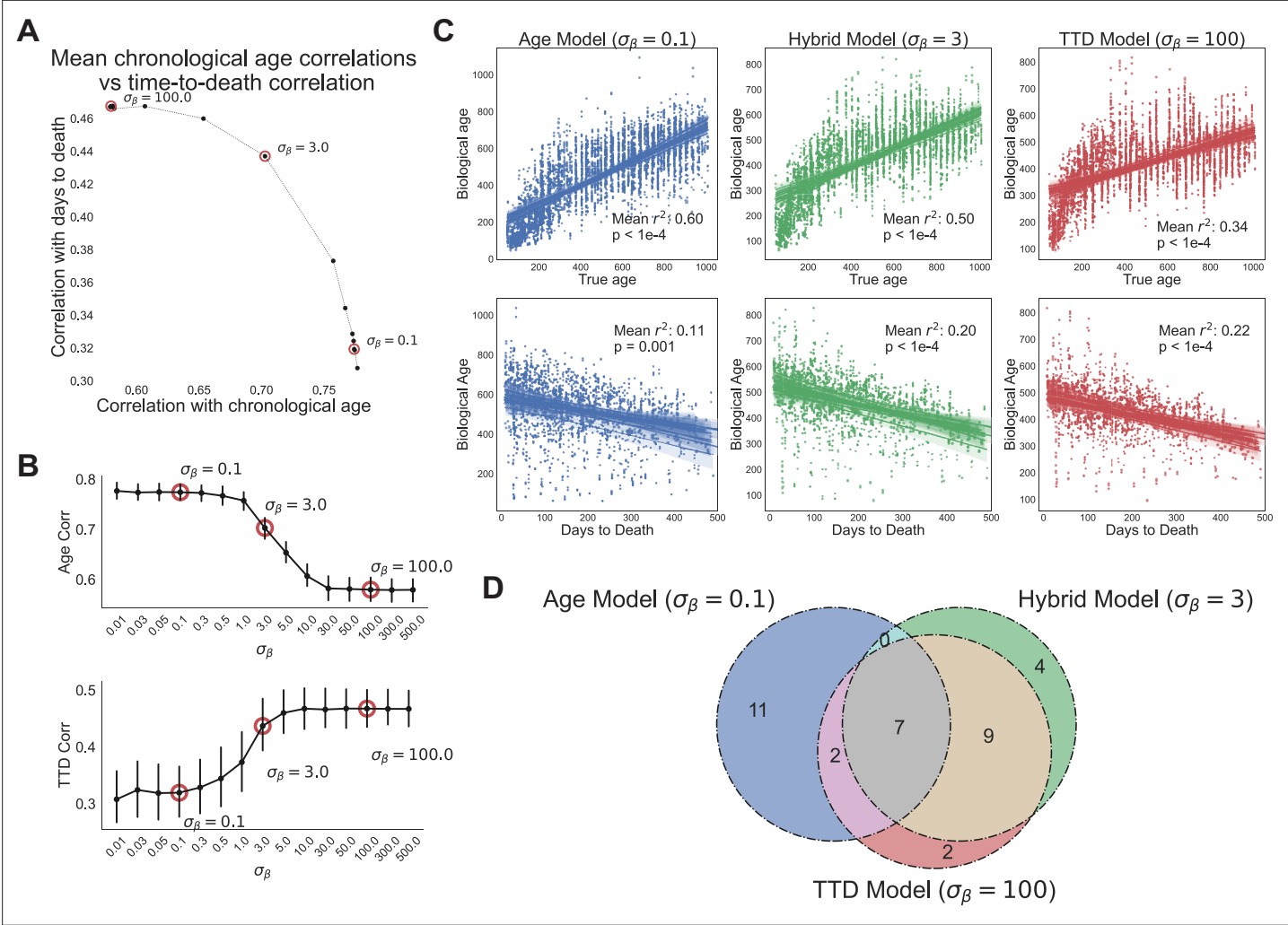

**Figure 4.** Age and time to death can be simultaneously predicted from physiological data. (**A**) Correlation coefficient of CASPAR biological age predictions (test set) with time to death (y-axis) and chronological age (x-axis) for different values of $\sigma_\beta$, ranging from 0.1 to 100. Circled points represent $\sigma_\beta$ values chosen for panels C and D. Correlation coefficients for each value of $\sigma_\beta$ represents the average of 15 models trained against independent training/test data splits. (**B**) Correlation coefficient of biological age predictions (test set) with chronological age (top) and time to death (bottom) versus $\sigma_\beta$ values. Error bars are SEM from the 15 models trained against independent training/test splits. (**C**) Scatterplots of predicted biological age versus chronological age (top) and time to death (bottom) for three different model classes (three different values of $\sigma_\beta$: 0.1, 3, 100). Points represent individual runs; all runs from each of the 15 independent test sets are shown. Lines represent linear regression for each of the 15 independent test sets. $R^2$ values are averages across the 15 test sets and p-values are $2\times$ of the median across the 15 test sets. (**D**) Venn diagram of the overlap between the top 20 most informative features for each of the three model classes noted above (age model, time to death model, hybrid model).

The online version of this article includes the following figure supplement(s) for figure 4:

**Figure supplement 1.** Age and time to death prediction.

activity (wheel running and pedestrian locomotion), and a surrogate of basal metabolic rate (VCO$_2$ while Sleeping). Outside of this overlap, quite a few features were distinct to either the age model or time to death model. Features more associated with time to death than with age ('TTD and Hybrid only' and 'TTD only') include measures of activity (wheel, food, and water intake) during the light phase or low activity states, perhaps indicating that a breakdown in circadian stability is an indicator of impending demise. Conversely, features more associated with age than time to death ('Age only') include the extent of activity during the dark phase, perhaps indicating that the best predictor of chronological age is a decline in high intensity activity. Limited overlap between age and mortality prediction models has been reported elsewhere using different measures of physiological age (*Schultz et al., 2020*), thus this may represent a more general truth about organism-level aging: aging-related

changes in physiological function, though of clear health relevance to the individual, are only partially related to mortality risk, demonstrating the need for a thoughtful balance between these conceptualizations of biological age.

## Discussion

Assessments of behavior and physiology are essential aspects of many preclinical studies. However, while technological advances such as sequencing have allowed researchers to explore molecular and cellular phenotypes in high-dimensional space via systems-level analyses, organism-level phenotyping has not benefited from a similar advance. To remedy this, we combined automated phenotyping cages with a sophisticated analysis pipeline to create a a platform for high-dimensional assessment of physiology and behavior in mice. This platform could be utilized to study multiple organism-level processes and diseases, for example, cognitive and mood disorders, neuromuscular deficits, or metabolic disease. We chose to focus on aging because (1) there is increasing interest in therapeutically modulating aging, (2) aging affects multiple physiological domains, so we expected broad and clear effects, and (3) current approaches to quantifying organism-level aging are extremely labor- and time-intensive (*Bellantuono et al., 2020*; *Sukoff Rizzo et al., 2018*). We were able to measure over 200 aging phenotypes across 22 physiological clusters, as well as network effects and related emergent phenomena (resilience) that are difficult to quantify via other methods, in >400 outbred mice. Data generation required ~20 hr of hands-on operator time per week for 1.5 years, that is, 0.75 FTE-years total. This study advances the state of the art for healthspan assessment in terms of throughput, resolution, and physiological scope.

At the outset of this study, it was not clear that monitoring of mice in a normal living environment would provide sufficient sensitivity to detect age-related changes. The rationale behind utilizing challenge-based assays is that animals must be pushed to the limit of their abilities in order to quantify functional decline (*Sukoff Rizzo et al., 2018*; *Bellantuono et al., 2020*). Although some aspects of aging undoubtedly require such assays, we found that our data show a plethora of aging-related changes in all age groups, including prior to 6 months of age, demonstrating that automated phenotyping of voluntary activity can detect even early aging-related changes. Mice reach sexual maturity at ~6 weeks of age, but are arguably still undergoing developmental changes for weeks or months after that. This shifts the question from when we *can* detect aging-related changes to when we *should*. Deciding whether aging-related changes in animals younger than 6 months of age represents aging, development, or a combination of the two is a complex issue that cannot be definitively resolved here. However, we did notice that phenotypes exhibited a diversity of trajectories across life – unchanged, parabolic, linear, logarithmic, and nearly exponential – and we propose that phenotypes which change monotonically, particularly when the trajectory is linear, can be considered part of aging even when that change begins early. For example, wheel running declines near-monotonically with age starting at 3 months; it seems reasonable to propose that this is an aspect of aging that begins quite early. Changes that are non-monotonic, for example, body weight, are more difficult to interpret through this lens.

One difficulty that arises when interpreting automated phenotyping data is distinguishing changes in physiology from changes in behavior. Although this is a somewhat arbitrary distinction, it is meaningful – a decline in overall energy expenditure because an animal runs less is different from a decline in energy expenditure due to reduced basal metabolic rate. To address this, we developed a robust HMM to assign a behavioral state to each 3 min time bin. We were then able to examine features conditioned on the state of the animal, for example, $VO_2$ while running. This turned out to be an informative approach, as state-conditioned features arising from the same sensor often clustered separately from one another, indicating that they contained complementary physiological information (e.g., pedestrian locomotion while eating versus while resting). More qualitatively, it allowed us to assess specific aspects of physiology that have long been considered the domain of specialized procedures. For example, energy expenditure while sleeping provides a reasonable surrogate of resting metabolic rate whereas $VO_2$ while running provides a reasonable surrogate of maximal oxygen consumption, both of which decline with age. We propose that behavioral inference based on automated phenotyping data is a useful technique that can be applied in a number of preclinical contexts.

A powerful use of multi-dimensional data is the application of network analysis to better understand the wiring of the system. We built a network of physiological and behavioral phenotypes using

sparse precision matrix estimation methods; in other words, we determined every pairwise correlation between features after accounting for all other features. In this context, two features connected by a strong edge (high covariance) are more likely to be mechanistically connected than two features with a weaker edge, and thus features that form a cluster are likely driven by the same mechanism(s), whereas features from a different cluster are likely driven by distinct mechanism(s). Aging involves a multitude of detrimental changes to health and well-being, but it is unknown how many distinct causal mechanisms drive these changes (*Freund, 2019*). Analyzing aging as a phenotypic network informs how many clusters exist, therefore how many independent mechanisms likely exist. The concept is analogous to identifying clusters of coordinately expressed genes; with sufficient data, one can conclude that coordinately expressed genes are regulated by a similar transcriptional program, and the number of gene clusters provides a reasonable estimate of the number of transcriptional programs. In this study, we identified 22 distinct organism-level clusters, though this number will undoubtedly be refined as additional studies are incorporated into the network analysis framework.

Network modeling also allows for quantification of overall network connectivity, which serves as a useful proxy for resilience. We uncovered striking changes with age in this dimension: more than virtually any individual feature, resilience smoothly and monotonically declined with age. The decline in resilience accelerated with age, decreasing more rapidly in old animals than in young animals, a pattern that is qualitatively consistent with the accelerating decline in health and corresponding exponential increase in mortality with age. This emergent property of the system was only detectable by analyzing the relationship between phenotypes, rather than the individual values of the phenotypes themselves – an analytical approach that is infeasible for data from challenge-based assays. Increasing network connectivity indicates that, with age, runs increasingly resemble one another, that is, old animals occupy a lower diversity of phenotypic states than younger animals. A similar phenomenon has been reported for human frailty – inter-individual variation in frailty scores decreases with age (*Rockwood et al., 2004*). It may be that individual animals become more physiologically inflexible with age; alternatively, this observation may be a consequence of survival bias, as DO mice reach 50% mortality by ~2 years of age, and only a small percentage of animals survive to >30 months. This latter explanation would imply that there are a limited number of viable aging trajectories, and animals that do not follow those trajectories die early, removing their contribution to phenotypic diversity. Longitudinal assessment of resilience in individuals would help address this question, and future work could develop measurements of individual animal resilience, rather than the population-level resilience we have calculated here. In particular, the second-by-second and minute-by-minute data streams from phenotyping cages could be used to examine how similar an animal is to itself at a later time. This 'temporal autocorrelation' analysis is only possible with time series data and has been proposed as a useful measure of individual resilience (*Scheffer et al., 2018*).

High-dimensional data is information-rich but also difficult to interpret. As such, there is value in developing summary statistics that capture meaningful aspects of the data. In the case of aging, this leads to the concept of biological age – a single number that is meant to reflect the aging-related health status of an animal better than does its chronological age. An increasingly popular method to quantify biological age from multi-dimensional data is to train a model to predict chronological age, then treat the error in that model (i.e., the difference between actual age and predicted age) as biologically meaningful. In many cases, this so-called 'delta age' correlates to meaningful outcomes, such as survival. Despite its popularity, the conceptual foundation of this approach is questionable. Consider that the computational goal of an age prediction model is to predict chronological age perfectly, but perfect performance would destroy the utility of the model. A model that predicts all 70-year-old people, healthy or unhealthy, to be exactly 70 provides no information other than chronological age itself; $R^2$ = 1 is not useful. However, although it is clear that perfect performance is not desirable, it is unclear how much error is optimal; that is, it is unclear whether the error should be 1%, 5%, 20%, or more. Further, the desire for error is not explicitly stated during model training – algorithms push toward perfect performance whenever possible and do not aim to capture any other type of information. At some point, improved age prediction begins to ignore the biological information we care about, but we do not know when that point is reached because we have not defined a 'ground truth' measurement of the variable we actually care about, that is, biological age. In short, if you don't know what you want, you're unlikely to get it. The non-circular approach is to define a measurement of biological age that is independent of the training data and use that as the outcome variable when

training models. However, because we don't fully understand aging, and because health is a difficult concept to define, let alone measure, it is difficult to confidently create such a 'ground truth' biological age metric.

A common alternative is to use lifespan, or survival time, as the outcome variable. Survival time is objectively measurable and fulfills the criteria of being independent of the training data, but it is still not exactly what we care about from an aging perspective: not all individuals with equal remaining lifespan are equally healthy or equally 'aged'. Lifespan is influenced by a large number of external factors, and mortality risk is often dominated by particular pathologies (e.g., cancer, in the case of many mouse strains). An anti-cancer drug or any other single-cause-of-death modulator does not meet most people's criteria of an aging intervention, but would be viewed as such if survival time were the single outcome variable assessed.

As neither chronological age prediction nor survival time prediction is an optimal method to evaluate biological age, we developed a compromise: CASPAR, which incorporates both chronological age and survival time prediction. CASPAR explicitly captures two key notions: (1) biological age should be correlated to chronological age and (2) deviations of biological age from chronological age should be informative of some proxy of health status (e.g., time to death). These two assumptions are often used as secondary validation metrics for mortality-only or age-only models, respectively: biological age predictions from mortality-trained models are tested for their correlation with chronological age, and biological age predictions from chronological-age models are tested for their ability to predict mortality. However, our approach explicitly quantifies these assumptions, allowing for smooth interpolation between chronological age regression and survival regression. We trained multiple models, varying the relative importance of age versus survival time prediction, and evaluated their performance on held-out animals. Age prediction was always more accurate than survival time prediction, but this is not surprising – the former is an exercise in estimating a concurrent variable (chronological age), whereas the latter predicts a future outcome that is dependent on a number of stochastic external influences that have not yet occurred and may have no relationship to an animal's current state. One extreme, the 'chronological age model', predicted chronological age well but had poor survival time prediction. Conversely, the 'time to death' model predicted survival time well, but showed substantially reduced age prediction performance. Therefore, both age and mortality information are present in the data, but the feature weighting that predicts them is largely distinct. A similar dichotomy between age and survival time prediction has been reported in other physiological datasets (*Schultz et al., 2020*; *Fischer et al., 2016*; *Levine et al., 2018*). Thus, considerable thought must go into model design, as different outcome variables are likely to emphasize different aspects of biology. We also demonstrate the existence of a hybrid model that gives some weight to both age and survival time prediction. This hybrid model predicted survival time nearly as well as a standard survival model, but gained substantial improvement in chronological age prediction. Notably, predicted biological age from the hybrid model was more strongly correlated to time to death than chronological age itself. It is beyond the scope of this manuscript to dictate exactly where the age versus survival prediction trade-off should lie, but we hypothesize that such hybrid models are likely to be more valuable and aging-relevant than either extreme.

The CASPAR approach also allowed us to take advantage of longitudinal measurements of the same mouse. Versions of the model in which we assumed that aging rates are relatively constant throughout life, thus allowing us to use multiple runs of the same animal to estimate a single age rate for that animal, performed significantly better than models without this additional constraint on held-out validation splits.

The model framework we have developed, with the ability to tune the relative weighting of age and survival time, can be applied more broadly. First, the underlying data does not need to be physiological. A similar approach can be used with any other high-dimensional, per-individual dataset, including methylation data, transcriptomic data, and blood biomarkers. For the purposes of preclinical intervention testing, we favor physiological data because it is, almost by definition, health-relevant. Our models relied on features like body mass, running speed, energy expenditure, and sleeping behavior – this is intuitively sensible and provides confidence that the resulting model truly reflects animal function. In contrast, although molecular data is often easier to acquire and contains more features, there is limited prior information on each individual feature, meaning that molecular data-based models are more difficult to interpret and sanity check. A second way to broaden the application of this modeling

framework is by using an outcome variable other than death. This may be particularly useful in human datasets, where follow-up mortality data is often limited. Age could be combined with a morbidity endpoint, or a pre-determined 'health score', and these outcome variables could be used to train the model. This may allow biological age assessment in cohorts with limited follow-up time.

Our study has several important limitations. First, it is not a fully longitudinal design. Mice were enrolled starting at different ages and followed for a fixed length of time (~1.5 years), thus our data are both left-truncated and right-censored. Having full lifespan curves for all animals would provide additional insight. Human clinical trial data has a similar left-censored, right-truncated data structure, so our ability to quantify health-relevant aging changes despite incomplete life histories bodes well for translationally relevant intervention testing using this platform. Second, the cohort was entirely female. This was a practical necessity; due to their aggressive tendencies, DO males are generally singly housed at all times, leading to unsustainable financial and vivarium space requirements. Nevertheless, a mixed dataset would be preferable, not only in terms of sex, but also in terms of strain and environmental variables, because mixed datasets increase the generalizability of conclusions (*Voelkl et al., 2020*, *Webster and Rutz, 2020*). It is quite likely that the specific physiological changes that develop with age will differ in male mice, in other strains, and in animals housed under different conditions; however, we suspect that broad patterns such as reduced wheel running, metabolic activity, and resilience, will generalize. In this manuscript, we have primarily focused on the analytical methodology and tools we developed to understand and summarize the physiological changes we observed because we believe those will be more generalizable, and thus useful, to the field than the particular set of physiological changes that we identify.

Additionally, although the automated monitoring cages we used provided a tremendous amount of data, they are blind to many health-relevant phenotypes such as body posture and coat condition. Additional high content, automated phenotyping modalities, such as video monitoring paired with machine vision feature extraction, would increase the value of this platform. Fourth, our most granular analyses used 3 min time windows. However, for many of the cage sensors, the data are acquired every second. This provides the opportunity for more sophisticated time series analyses that track individual behaviors and rapid fluctuations in physiology.

Changes with age occur at all levels of biological organization (molecules, cells, tissues, etc.). We chose to focus on physiology and behavior because changes to this level of biological organization are most proximal to changes in health and quality of life. We created a platform that can measure physiological and behavioral aging at any stage of life in outbred mice using automated phenotyping. This system provides a number of advances that allow organism-level aging to be studied with improved throughput, resolution, and physiological scope while reducing the activation energy that comes with highly specialized assay procedures. We encourage more widespread adoption of automated phenotyping and high-dimensional analysis in order to study aging and putative aging interventions.

## Materials and methods

### Animals

All experiments were conducted according to protocols approved by the Calico Institutional Animal Care and Use Committee, protocol numbers C-1-2015 and C-1-2017. Female DO mice were obtained from the Jackson Laboratory (Bar Harbor, ME), and housed at Calico in ventilated caging with a 12 hr light cycle and ad libitum access to food and water. Mice were group housed when not being monitored in phenotyping cages. As mice died over the course the experiment, new mice were enrolled to maintain ~250 mice on study at any given time. Cohort size was initially targeted at >300 animals based on known variation in DO mouse phenotypes (e.g., body weight) and lifespan variation, though no specific power analysis was employed.

### Data collection

Mice were profiled in Promethion High-Definition Multiplexed Respirometry Cages (from Sable Systems International) on a 1 week on, 3 weeks off cycle until death or study termination (~1.5 years after study initiation). Mice were singly housed when in Promethion cages and had ad libitum access to food and water.

Each cage records raw measurements from seven integrated sensors at 1 s resolution: water bottle mass module, food hopper mass module, sleeping chamber mass module (for mouse weight), wheel sensor, and X/Y/Z infrared beams. Each cage also records gas measurements from an oxygen sensor, carbon dioxide sensor, and humidity sensor at 3 min resolution.

From these raw data, a data processing macro provided by Sable Systems was used to generate features at 3 min interval: $VO_2$, $VCO_2$, $VH_2O$, energy expenditure, respiratory quotient, food and water intake, walking speed, all movement, wheel speed, X/Y/Z beam breaks.

## Data analysis

High-level summary: Following initial macro processing, the data were analyzed in four main stages. First, outlier detection was run to remove outlier points and identify instrument failures. Next, this processed data was used to train an HMM which was used to identify distinct physiological states and assign each 3 min timepoint to one of these states. After state assignment, we derive 309 per-run features from the combination of the cage measurements and inferred physiological states. We use the term 'per run' to indicate that we treat each run as an independent event and take the average of each feature over the time in that run. Finally, we use network analysis to identify aging-related features, cluster features into phenotype clusters, assess resilience, and train models.

Data and code is available at https://github.com/calico/catnap, (copy archived at swh:1:rev:2f18ee-a0d02c23501bdd36558822a6974f99f640; *Chen, 2002*) including Python scripts to train and run the CASPAR model (*Chen et al., 2021*).

## Outlier detection and QC

Outlier detection was performed on each channel of a run (a week in the metabolic cage) independently. To eliminate the stress of acclimatizing to the metabolic cage as a potential confounder, we dropped the first 24 hr of data (empirically we observed that mice reached a steady state after the first light cycle). Runs shorter than a full day (after truncation) were removed to avoid bias. We removed 77 runs for being too short (2.2%).

We then performed a range check on each channel, censoring measurements that were outside of plausible physiological range (0–10 for gas measurements, 0–100 for walking and wheel speed, 5–80 g for body mass, and 0–2000 for beam breaks). These extremely permissive ranges were adopted to detect sensor faults.

We next identified outliers in the five gas measurements by removing the circadian component of each channel using RobustSTL (*Wen et al., 2019*) and applying a generalized extreme studentized deviate (ESD) test for outliers after subtracting the circadian trend. We used 24 hr as the period for RobustSTL seasonality extraction and 0.05 significance level for the ESD test.

To avoid low power in detecting outliers from multiple hypothesis correction, we set the upper bound for the generalized ESD test to 30% of the number of data points in a run and removed a run if 30% or more of its points were flagged as outliers. We removed 295 runs (~8.4%) for having too many outlier timepoints.

Data points which were flagged as outliers were censored, thus, no imputation of any data points was done.

## HMM and state assignment

In order to identify common activity states and assign timepoints to these states, we trained a discrete state HMM on the data after QC and outlier detection. We extend the standard discrete HMM, leveraging the multi-dimensional nature and the dense temporal sampling of our data to identify unreliable measurements. First, to account for intermittent sensor failure and to identify subtle outliers that were not detected in the outlier detection and QC stage, we augmented the HMM with two additional states: (1) a 'censored' state which models spurious zero readings from the sensors and (2) a 'noise' state which models measurements that are unlikely to be observed given other measurements at the same timepoint. Second, to model batch effects due to calibration of gas sensors and other cage-related biases, we learn a gas analyzer-specific batch correction offset per channel which was fit simultaneously with the HMM parameters.

In order to determine the number of HMM activity states, we split the dataset into a training set of 2689 runs and a held-out a validation set of 471 runs. We then trained our robust HMM on the training

set varying the number of latent states between 1 and 10 and determined the number of latent states using the one standard error rule on the log-likelihood on the held-out validation dataset. This selection process yielded 6 as the optimal number of states, we then fit a final robust HMM using all the data with 6 states.

Labels were then assigned to each of the states identified by the robust HMM based on the distribution of the underlying 14 raw cage measurements conditioned on each HMM state.

## Features

We controlled for exposure to the phenotyping cage as a confounder using ANOVA/multiple linear regression. After introducing a new variable 'run number' for the number of times a mouse has been profiled in the phenotyping cages, we fit a regression model regressing out the effect of 'run number' and interactions between 'run number' and the HMM state on all measurements. This allowed us to learn a correction for exposure effects specific to each state for each measurement.

To enable easier comparison across ages, we also normalized the gas measurements for body mass. We fit a multiple linear regression model regressing each gas measurement on body mass and interactions between body mass and the HMM state. We then normalized each gas measurement to the mean body mass across all runs (31 g).

All regressions were performed using algorithms provided in the Scikit-learn package. We derived 309 aggregate features from each run (excluding partial days) including:

- Means of base measurement across the entire run.
- State occupancy across the entire run.
- Means of base measurements during each state across the entire run
- Means of base measurement and state occupancy in 4 hr periods aligned to the light cycle.
- Ratio of base measurement and state occupancy pre/post light transitions.
- Frequency, duration, and interval between bouts of feeding, exercise, and sleep. Bouts were determined from individual 3 min data streams rather than HMM states, as the latter do not reflect single behaviors.

In all results shown, figures indicate means across runs of these features with the standard error of the mean as error bars. p-Values with respect to age were calculated via univariate KT test and Bonferroni-adjusted for multiple hypothesis testing.

## $\ell$-1 trend filtering

In order to account for run-to-run batch variability, we apply $\ell - 1$ trend filtering (**Kim et al., 2009**) to mice with $\geq 4$ runs individually, treating each mouse as a 309-dimensional time series with 1 timepoint per run. Features were standardized to zero mean and unit variance and smoothed using multivariate $\ell - 1$ trend filtering with an $\ell - 1\ell - 2$ penalty (and then re-scaled back to their original scales). The $\ell - 1\ell - 2$ penalty captures our intuition that real, dramatic, changes in animal physiology should be captured across multiple sensor modalities or derived features (or conversely that abrupt changes in a single or small number of features are more likely to be noise). The $\ell - 1$ trend filtered data is only used for the subsequent analysis steps (aging rate regression and constructing phenotype networks).

## Aging rate regression model

We fit aging rate regression models using the 309 features described above. In contrast to the typical regression models on chronological age or survival, our aging rate regression framework allows us to both (1) leverage repeated longitudinal measurements of the same mouse and (2) incorporate both age and survival regression into a unified framework. In order to evaluate the performance of the aging rate regression model and to investigate the impact of various assumptions in the model, we train models on 15 random animal hold-out splits – holding out all runs of 10% of mice to form a test set and training the models on the remaining 90% of mice – and report mean $R^2$ statistics for age and time to death predictions, averaged across all random splits. p-Values when reported are the $2\times$ the median p-value over all random splits as a conservative bound on the p-value across multiple data splits (**Romano and DiCiccio, 2019**).

At a high level, the aging rate regression model assumes that individual animals have an 'aging-rate' $\beta$ which uniformly speeds up/slows down their rate of aging relative to some reference animal. We infer this aging rate by comparing a proxy of animal health (remaining lifespan) to the reference

animal – as a concrete example, a 12-month-old mouse with $\beta = 1.5$ would have the same expected remaining lifespan of a $12 \times 1.5 = 18$-month-old 'reference mouse'. We infer the aging rate by extending the classical accelerated failure time model (**Wei, 1992**) commonly used in survival analysis. Modeling the time to death of an animal by

$$\log Y_i^j = \beta_i^j + Z_i^j$$

where $Y_i^j$ is the time to death of mouse $j$ at run , $\beta_i^j$ is the inferred $\beta$ for mouse $j$ at run , and $Z_i^j$ the random variable denoting the remaining lifespan of a reference mouse at the predicted age for mouse $j$ at run . In order to estimate this aging rate, we must first estimate $Z$ which we do by fitting a left-truncated, right-censored log extreme value distribution to estimate the distribution of DO mouse lifespans and use that to compute the distribution of remaining lifespan for a mouse of a given age.

In order to incorporate multiple measurements of the same animal at different ages, we additionally assume that each mouse has some latent aging rate $\beta^j \sim N(1, \sigma_\beta^2)$ and that each observation of a mouse has observed $\beta_i^j \sim \beta^j + \epsilon$ for $\epsilon \sim N(0, \sigma_\epsilon^2)$. In practice, we marginalize out the $\beta_i^j$, leaving $\sigma_\beta^2$ and $\sigma_\epsilon^2$ as hyper-parameters to be determined.

$\sigma_\epsilon^2$ represents our prior on the variance of the aging rate across different observations of the same mouse, a small value denoting a strong prior for a single consistent $\beta$ over the entire lifespan of a mouse. We find that small values of $\sigma_\epsilon^2$ generally outperform larger values on a held-out validation set which supports our assumption of a component of aging rate that is uniform throughout life. However, the model was not sensitive to small changes in value of $\sigma_\epsilon^2$ and in all subsequent experiments and results, we set $\sigma_\epsilon^2 = 1$.

Similarly, $\sigma_\beta^2$ denotes our prior on the variance of the aging rate within the population. When $\sigma_\beta^2$ is set to a small value, it represents a belief that all animals age at roughly the same rate. In this regime, the aging rate regression model approximates chronological age regression (mostly ignoring information about remaining lifespan). When $\sigma_\beta^2$ is large, it represents a belief that animals can age at very different rates, so the aging rate regression model emphasizes the health component (in this case, time to death) and approximates an accelerated failure time survival regression model (largely ignoring the chronological age of the animal). By adjusting $\sigma_\beta^2$, we can fit hybrid models that take both into account.

We fit the aging rate regression models by maximizing the likelihood:

$$P(T_i^j \mid age = F(X_i^j)) \cdot \int P(\hat{\beta}_i^j \mid \beta^j) \cdot P(\hat{\beta}^j) d\hat{\beta}_i^j$$

where $A_i^j$ is the age of mouse $j$ at run , $\hat{\beta}_i^j = \frac{F(X_i^j)}{A_i^j}$, $T_i^j = \log Y_i^j - \hat{\beta}_i^j$, $\hat{\beta}^j = \text{mean}_i \hat{\beta}_i^j$ and $F(X_i^j)$ is the predicted biological age of features $X_i^j$ of mouse $j$ at run . In our experiments, we implemented the aging rate regression models using gradient boosted decision trees for $F(\cdot)$ (**Chen and Guestrin, 2016**).

## Age-specific phenotype networks

We fit age-specific phenotype networks using taking 3-month age bins starting from age 0 as separate age bins. The graphical LASSO is a method of fitting a sparse Gaussian graphical model – allowing us to identify putative causal relationships between features. In order to account for changes in relationships between features as mice age, we used a variant of the graphical LASSO called time-varying graphical LASSO (**Hallac et al., 2017**) which fits a separate model per age bin but penalizes differences between adjacent age bins (to encode our prior that changes in network connectivity occur gradually with age). We used the standard $\ell - 1$ sparsity penalty per age bin and a perturbed node penalty between age bins (**Mohan et al., 2012**). Since several of our features are non-Gaussian in distribution, we adopted the nonparanormal covariance estimator described in **Liu et al., 2009**.

In order to cluster features into phenotype clusters, we bootstrap sample the estimated graphical model 1000 times and consensus cluster (**Monti et al., 2003**) features using spectral clustering. The number of clusters to use was determined using the approach described in **Monti et al., 2003**. We estimate the degree of interdependence between features using the negative copula entropy (log-determinant of the estimated covariance matrix) (**Singh and Póczos, 2017**) which we termed 'resilience'.

## Heritability and genetic correlations

We collected tail clippings and extracted DNA from all animals. Samples were genotyped using the 143,259-probe GigaMUGA array from the Illumina Infinium II platform (*Morgan et al., 2015*) by NeoGen Corp. (genomics.neogen.com/). We evaluated genotype quality using the R package: qtl2 (*Broman et al., 2019*). We processed all raw genotype data with a corrected physical map of the GigaMUGA array probes (https://kbroman.org/MUGAarrays/muga_annotations.html). After filtering genetic markers for uniquely mapped probes, genotype quality, and a 20% genotype missingness threshold, our dataset contained 116,377 markers.

For each mouse, starting with its genotypes at the 116,377 markers and the genotypes of the eight founder strains at the same markers, we inferred the founders-of-origin for each of the alleles at each marker using the R package: qtl2 (*Broman et al., 2019*). This allowed us to test directly for association between founder-of-origin and phenotype (rather than allele dosage and phenotype, as is commonly done in QTL mapping), and used these founder-of-origin inferences to compute the kinship between pairs of mice for heritability and genetic correlation analyses.

For each of the 309 derived phenotypes in this study, we computed the heritability (proportion of phenotypic variance explained by additive genetic effects, or PVE) using a custom implementation of EMMA (*Kang et al., 2008*), a standard linear mixed model used for genetic association analyses. Specifically, for each mouse, we randomly sampled a representative run to construct a 415 × 309 matrix of phenotype values, with each column quantile-normalized to the standard normal distribution. We computed heritability for each phenotype while controlling for fixed effects of age and cohort (since the runs spanned the entire age and cohort distribution in the study). This workflow was repeated for 100 random draws of runs for each mouse, and the median (and inter-quartile range) of the estimated heritability was reported for each phenotype.

For the 45 phenotypes with significant nonzero heritability, we compute genetic correlation for each pair of phenotypes, using a matrix-variate linear mixed model (*Furlotte and Eskin, 2015*), while conditioning on the fixed effects of age and cohort. Similarly, we computed the partial phenotypic correlation for each pair of phenotypes, controlling for age and cohort effects.

## Acknowledgements

We thank Kevin Wright for assistance processing genotype data. We thank Sable Systems for building the custom system and for continual consultation on hardware, software, troubleshooting, and analysis; particular thanks to John Lighton, Thomas Foerster, and Jeffrey Richardson.

## Additional information

### Competing interests

Zhenghao Chen, Anil Raj, GV Prateek, Andrea Di Francesco, Justin Liu, Ganesh Kolumam: is affiliated with Calico Life Sciences LLC. The author has no financial interests to declare. Brice E Keyes, Vladimir Jojic, Adam Freund: was affiliated with Calico Life Sciences LLC during the time of his contribution to the study. The author has no financial interests to declare.

### Funding

| Funder | Grant reference number | Author |
| --- | --- | --- |
| Calico Life Sciences, LLC | | Zhenghao Chen<br>Anil Raj<br>GV Prateek<br>Andrea Di Francesco<br>Justin Liu<br>Brice E Keyes<br>Ganesh Kolumam<br>Vladimir Jojic<br>Adam Freund |

The funders had no role in study design, data collection and interpretation, or the decision to submit the work for publication.

## Author contributions
Zhenghao Chen, Anil Raj, Data curation, Formal analysis, Methodology, Software, Visualization, Writing – original draft, Writing – review and editing; GV Prateek, Data curation, Formal analysis, Methodology, Software, Writing – original draft, Writing – review and editing; Andrea Di Francesco, Investigation, Methodology, Resources, Writing – original draft, Writing – review and editing; Justin Liu, Brice E Keyes, Investigation, Methodology, Resources; Ganesh Kolumam, Supervision; Vladimir Jojic, Data curation, Formal analysis, Methodology, Software, Supervision, Writing – original draft, Writing – review and editing; Adam Freund, Conceptualization, Formal analysis, Funding acquisition, Project administration, Resources, Supervision, Writing – original draft, Writing – review and editing

## Author ORCIDs
Anil Raj ⬤ http://orcid.org/0000-0003-4412-0883
Andrea Di Francesco ⬤ http://orcid.org/0000-0001-6867-8203
Justin Liu ⬤ http://orcid.org/0000-0002-5338-6491
Adam Freund ⬤ http://orcid.org/0000-0002-7956-5332

## Ethics
All experiments were conducted according to protocols approved by the Calico Institutional Animal Care and Use Committee, protocol numbers C-1-2015 and C-1-2017.

## Decision letter and Author response
Decision letter https://doi.org/10.7554/eLife.72664.sa1
Author response https://doi.org/10.7554/eLife.72664.sa2

## Additional files

### Supplementary files
• Supplementary file 1. Univariate Kolmogorov-Smirnov test for each feature against age. Rows are each of the 309 per-run features, columns are the KT coefficient, p-value with respect to age, Bonferroni-adjusted p-value, and a binary TRUE/FALSE significance column to indicate whether the adjusted p-value is less than 0.05. Percent of features with a significant age association is indicated on the right side of the table.

• Supplementary file 2. Features that comprise each of the clusters identified in *Figure 3a*. Each column represents 1 of the 22 identified clusters. Row 1 indicates cluster number, row 2 indicates the qualitative name assigned to that cluster, and subsequent rows list the features that are included in that cluster.

• Transparent reporting form

### Data availability
All data and associated code are available on Github (https://github.com/calico/catnap, (copy archived at swh:1:rev:2f18eea0d02c23501bdd36558822a6974f99f640)).

The following dataset was generated:

| Author(s) | Year | Dataset title | Dataset URL | Database and Identifier |
|---|---|---|---|---|
| Chen Z, Raj A, Prateek GV, Francesco AD, Liu J, Keyes BE, Kolumam G, Jojic V, Freund A | 2022 | Automated, high-dimensional evaluation of physiological aging and resilience in outbred mice | https://github.com/calico/catnap | Github, Github |

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
