## [Editor Report]

Chen et al., develop a comprehensive platform to score aging-dependent changes in mouse physiology and behavior using a multi-dimensional longitudinal phenotyping approach. Their thorough data collection and analysis reveals a diversity of trajectories in aging-related physiological and behavioral changes and helps disentangle biological aging from chronological aging, providing a pioneering reference for future studies aimed at large-scale aging multi-dimensional phenotyping.

---

## [Decision Letter]

**Decision letter after peer review:**

Thank you for submitting your article "Automated, high-dimensional evaluation of physiological aging and resilience in outbred mice" for consideration by *eLife*. Your article has been reviewed by 2 peer reviewers, including Dario Riccardo Valenzano as Reviewing Editor and Reviewer #1, and the evaluation has been overseen by Christian Rutz as the Senior Editor.

Essential revisions:

1) Explain the feature normalization approach and clarify how it influences their many feature covariance analyses.

2) Elaborate, comment or revise the operative definition of "resilience" (see reviewer #2's comment on resilience).

3) Explain the interpretation of the hyperparameter in the CASPAR model.

4) Please expand your data availability statement and ensure that the data and code are publicly available.

5) Address in the rebuttal letter all the comments from the reviewers.

*Reviewer #1 (Recommendations for the authors):*

If this wants to be a "methods" paper, rather than a "novel finding" work, as clearly indicated by the choice of the title, then the authors should make sure their methods and analyses are available, intelligible and easy applicable to other researchers. Can the authors provide evidence that even larger-scale monitoring of mouse behavior/physiology published would benefit from their analytical approach?

Alternatively, if the authors choose to focus more on the results that emerge from their analysis (question-driven approach), I would suggest to highlight more the novelty of their findings in the context of the current understanding of aging in unchallenged conditions and possibly rephrase the title to reflect the findings, rather than only their approach.

*Reviewer #2 (Recommendations for the authors):*

I describe all major points in the public review.

---

## [Author Response]

Essential revisions:1) Explain the feature normalization approach and clarify how it influences their many feature covariance analyses.

We apologize if this methodology was unclear. First, all measures were corrected for exposure to the metabolic cage using a linear model, using run number as a covariate. Additionally, after exposure correction, gas measurements (VO2, VCO2, VH2O, EE) were corrected for body mass. The corrected values were used for all analyses (Figures 2-4). Please see our response to reviewer #2 for more detail.

2) Elaborate, comment or revise the operative definition of "resilience" (see reviewer #2's comment on resilience).

There is no gold-standard measure of organismal resilience, and there are several reasons why we believe the metric we employ here is, at the very least, a valuable addition to the resilience toolbox. First, we believe our metric does capture the ability of the system to respond to change. Second, unlike utilizing a specific perturbation, our metric incorporates many dimensions of physiology. Third, only specific types of physiological changes (those that alter the overall network connectivity score) will affect our measurement of resilience. Please see our response to reviewer #2 for more detail.

3) Explain the interpretation of the hyperparameter in the CASPAR model.

The sigma_beta hyperparameter is a metric that quantifies the model’s assumption that different individuals age at different rates. Thus, it is a reflection of the researcher’s belief in the variability of aging rate between individuals. If a ground truth measurement of individual aging rates existed, the value of this hyperparameter could be measured empirically. Unfortunately, there remains no agreed-upon definition of biological age and no such ground truth is available. Therefore, rather than a priori choosing a single value of sigma_beta, we explore model behavior under different values of sigma_beta. This allows us to assess versions of the model that avoid overfitting to either chronological age or time to death, both of which are imperfect biological age surrogates. We regret that our description was unclear, and we have edited the text to clarify and further describe the meaning of the sigma_beta parameter. Please see our response to reviewer #2 for more detail.

4) Please expand your data availability statement and ensure that the data and code are publicly available.

We have expanded this statement and ensured that all data and code are available on Github (https://github.com/calico/catnap)

5) Address in the rebuttal letter all the comments from the reviewers.

Please see below.

Reviewer #1 (Recommendations for the authors):If this wants to be a "methods" paper, rather than a "novel finding" work, as clearly indicated by the choice of the title, then the authors should make sure their methods and analyses are available, intelligible and easy applicable to other researchers. Can the authors provide evidence that even larger-scale monitoring of mouse behavior/physiology published would benefit from their analytical approach?Alternatively, if the authors choose to focus more on the results that emerge from their analysis (question-driven approach), I would suggest to highlight more the novelty of their findings in the context of the current understanding of aging in unchallenged conditions and possibly rephrase the title to reflect the findings, rather than only their approach.

We agree with the reviewer that this study could be framed in multiple ways. We chose to frame the narrative around the methodology and analyses because we believe those will be more useful to the field than the particular set of physiological changes that we identify, though these are also interesting.

Initially we, and many of our colleagues, were unsure whether our platform would have sufficient sensitivity to detect physiological and behavioral changes with age. Thus, in our opinion, the most impactful aspect of this study is not the list of individual features we identified, but the demonstration that the approach is viable: passive, automated monitoring provides a rich, multi-dimensional view of physiological aging, and does so in outbred mice (a particularly high bar). We also provide a set of analytical tools to facilitate interpretation of this kind of multi-dimensional, organism-level data, i.e. annotating behavioral states, quantifying systems-level connectivity and simultaneously predicting chronological age and remaining lifespan.

This proof of concept paves the way for larger and more complex studies. It is now feasible to (1) scale the system, (2) incorporate additional passive monitoring technologies (e.g. video), (3) run fully longitudinal studies, (4) incorporate both sexes, (5) test many interventions simultaneously, (6) avoid the idiosyncrasies of inbred strains, and (7) acquire actionable results (e.g. the intervention works or it doesn’t) within months. We also expect that the study of other disease processes affecting behavior and physiology could benefit from this approach. To facilitate these future studies, we have made all of our data and code publicly available.